# Antiviral Cell Products against COVID-19: Learning Lessons from Previous Research in Anti-Infective Cell-Based Agents

**DOI:** 10.3390/biomedicines10040868

**Published:** 2022-04-07

**Authors:** Irina Chikileva, Irina Shubina, Anzhelika-Mariia Burtseva, Kirill Kirgizov, Nara Stepanyan, Svetlana Varfolomeeva, Mikhail Kiselevskiy

**Affiliations:** 1Research Institute of Experimental Therapy and Diagnostics of Tumor, NN Blokhin National Medical Center of Oncology, 115478 Moscow, Russia; irinashubina@mail.ru (I.S.); kisele@inbox.ru (M.K.); 2College of New Materials and Nanotechnologies, National University of Science and Technology “MISiS”, 119049 Moscow, Russia; m2003567@edu.misis.ru; 3Research Institute of Children Oncology and Hematology, NN Blokhin National Medical Center of Oncology, 115478 Moscow, Russia; kirgiz-off@yandex.ru (K.K.); nara19922@yandex.ru (N.S.); varfolomeeva-07@mail.ru (S.V.)

**Keywords:** COVID19, CAR-lymphocytes, vaccines, dendritic cells, antiviral effectors, natural killer cells, cytokine storm, immunoglobulin G, biotechnology

## Abstract

COVID-19 is a real challenge for the protective immunity. Some people do not respond to vaccination by acquiring an appropriate immunological memory. The risk groups for this particular infection such as the elderly and people with compromised immunity (cancer patients, pregnant women, etc.) have the most serious problems in developing an adequate immune response. Therefore, dendritic cell (DC) vaccines that are loaded ex vivo with SARS-CoV-2 antigens in the optimal conditions are promising for immunization. Lymphocyte effector cells with chimeric antigen receptor (CAR lymphocytes) are currently used mainly as anti-tumor treatment. Before 2020, few studies on the antiviral CAR lymphocytes were reported, but since the outbreak of SARS-CoV-2 the number of such studies has increased. The basis for CARs against SARS-CoV-2 were several virus-specific neutralizing monoclonal antibodies. We propose a similar, but basically novel and more universal approach. The extracellular domain of the immunoglobulin G receptors will be used as the CAR receptor domain. The specificity of the CAR will be determined by the antibodies, which it has bound. Therefore, such CAR lymphocytes are highly universal and have functional activity against any infectious agents that have protective antibodies binding to a foreign surface antigen on the infected cells.

## 1. Introduction

SARS-CoV-2 (severe acute respiratory syndrome coronavirus 2) is a single-stranded RNA virus, infectious agent of coronavirus disease 2019 (COVID-19). It causes viral disease with highly variable symptoms, which range from asymptomatic condition to acute respiratory distress syndrome with possibility of neurological damage, and even death [1,2]. Current treatment strategies include anti-viral drugs such as remdesivir and ribavirin, convalescent plasma, anti-IL-6 receptor antibodies, and corticosteroids to dampen the inflammatory immune response and are reviewed elsewhere [1,3]. Several effective vaccines against SARS-CoV-2 have been rapidly developed all over the world by global clinical investigators since the beginning of the epidemic [4,5]. However, certain patients do not develop a desirable immune response after the immunization [4]. Actually, the vaccines efficacy ranges between 60–95%, depending on their type and way of administration [4]. The introduction of new viral strains diminishes the efficiency of the vaccination. Most importantly, persons with immune disorders who are at the highest risk to severely suffer from the SARS-CoV-2 infection have lower potential to develop resistance when immunized [6]. The risk of low immunogenicity of COVID-19 vaccines is especially high in solid organ transplant recipients and patients with hematological malignancy [6]. The second problem of this severe disease is that in case of developing an immune response to the natural infection where the acquired protective immunity does not last long, for about six months. SARSCoV2 was shown to suppress functions of T-cells and dendritic cells (DCs) [7]. Thus, enhancement of the immune response is highly desirable in such cases. DCs are the most potent antigen-presenting cells (APCs). So far, DC vaccines have been used mainly in oncology with well-established safety [8]. Habitually, DCs for tumor vaccines are generated from autologous monocytes (moDCs) in the presence of recombinant granulocyte-macrophage colony-stimulating factor (GM-CSF) and interleukin-4 (IL-4). Tumor-associated antigens (TAA) usually possess low potential to induce an immune response. However, DCs that are generated ex vivo and loaded with TAA may overcome this obstacle. Multiple experiments proved their high potential in experimental animal models of tumor treatment [9]. Nonetheless, their efficacy as cancer monotherapy is still insufficient [8,10,11]. The low potential of antitumor DC vaccines is determined by a number of factors including vaccine treatment for advanced cancer, practically for incurable patients, although vaccination has the greatest effect as prophylaxis, which was perfectly demonstrated by the high effectiveness of experimental antitumor DC vaccines in animals. Other factors that have impact on low DC vaccine effectiveness involve tumor heterogeneity, low immunogenicity of tumor antigens, and a lack of significant differences from normal tissues. Contrastingly, SARS-CoV-2 is a virus with its proprietary antigens. Anti-viral DC-vaccines may be used for prophylaxis and, therefore, are anticipated to have high efficacy alongside with safety. Most importantly and intriguingly, DCs may be manipulated in vitro to induce different types of immune response or even suppress undesirable autoimmune reactions [12,13]. That makes them extremely attractable as the possibility to achieve a desirable immune response against SARS-CoV-2, as neither natural infection nor current vaccines induce a robust and durable immune response. The immune response should be simultaneously moderate enough not to induce a cytokine storm, which is especially dangerous during the infection. DC vaccines that are loaded ex vivo with SARS-CoV-2 antigens in the optimal conditions could have a vital potential for the immunization of people with problems in the development of protective immune reactions that might be unable to mount any response to vaccines. 

However, anti-viral DC-vaccines, just as conventional vaccines, are intended for prophylaxis. There are situations, which require immediate assistance to an organism that is incapable of managing a viral or another challenge by its own forces. Antibodies from the immune organisms such as antivenoms might act to save weaker patients from a potentially deadly threat such as tetanus toxoid or a snakebite. Actually, anti-SARS-CoV-2 antibodies are the main acting substances of convalescent plasma, derived from the patients who recovered from SARS-CoV-2. Their plasma is employed as a last resort for critically ill patients with COVID19 [3,14]. At the disease onset, SARS-CoV-2 induces antibodies that act in neutralizing the virus. Subsequently, the antibodies are switched to immunoglobulin (Ig) G-type, which is capable of engaging antibody-dependent cell-mediated cytotoxicity (ADCC) and antibody-dependent endocytosis (ADEn). NK-cells possess low affinity receptor of IgG (FcγRIII) CD16 that allows them to detect via antibodies and destroy cells with viral components on their membranes. This process is termed ADCC. ADEn is executed by macrophages, monocytes, and neutrophils, which recognize through Fcɣ receptors such as CD16 and CD64 (FcγRI) viral particles that are complexed with IgG and endocytose them. Notably, although robust IgM and IgA responses were developed in both survivors and non-survivors with severe COVID-19, non-survivors showed attenuated IgG responses, accompanied by compromised Fcɣ receptor binding and Fc effector activity [15,16]. That clearly indicates the important role of ADCC in clearance of the virus. Chimeric antigen receptor (CAR) combines the properties of antibodies and T-cell receptors (TCR). Its receptor part consists of a modified antigen-binding antibody fragment, so-called single-chain variable fragment (scFv). scFv is linked via a hinge and a transmembrane domain with signaling moieties of TCR (CD3ζ) and its co-stimulators such as CD28, OX-40, and 4-1BB [17]. First-generation CARs contained only CD3ζ stimulatory moieties [18]. However, their activating potential was not optimal. Pule et al. improved CAR-construct by replacing CD3ζ-transmembrane domain with CD28 analogous and adding CD28-costimulatory domain to CD3ζ [19]. Notably, CD28 co-stimulation increases IL-2-independent proliferation and enhances the resistance of CAR-T cells to T regulatory cells [20]. Third-generation CARs include three activating domains. Besides CD28, other domains such as 4-1BB and OX40 that additionally favor survival and activation of CAR-T cells are added [21,22]. Certain data indicate that second-generation CAR-T cells that are based on CD28 are highly susceptible to exhaustion [22]. 

Other signaling moieties may be applied as well. CARs recognize an antigen without the need for the presentation on the major histocompatibility (MHC) molecules. That is the property that is rendered by the modified antibody part-scFv. Effector CAR-lymphocytes provide high cytotoxic activity upon antigen recognition. Thus, their mechanism of action resembles ADCC. The studies showed that CAR-T cells were highly effective in the treatment of a number of hematological neoplasms, especially in acute lymphoblastic leukemia (ALL), with complete remission in 90% of patients [23]. Genetically engineered CAR-lymphocytes are being investigated for therapy of multiple solid tumors.

About a decade ago, another type of CARs termed universal was also developed [24]. Such CARs are based on molecules that bind natural or modified TAA-specific antibodies. The receptor part of such CARs may be presented by Fc-receptors such as CD16 or CD64, avidin that bind biotinylated antibodies, etc. Such CAR-lymphocytes clearly provide ADCC towards tumor cells opsonized by antibodies similar to NK-cells. However, CD16 on NK-cells is tightly regulated by a proteolytic process that rapidly downregulates its expression upon NK cell activation [25]. Engineered CARs may lack that negative feedback loop and are predicted to act more potently [26]. 

Intensive studies have started to introduce CAR technology for COVID-19 treatment, although previously it was used primarily in oncology.

This review focuses on different biomedical cell products (BMCP) that may target SARS-CoV-2, namely DC vaccines and CAR-lymphocytes. The potential of universal CAR-lymphocytes in the COVID-19 treatment is discussed. 

## 2. Promising BMCP in COVID-19 Prophylaxis and Treatment

### 2.1. DC Vaccines 

Nowadays, a few anti-SARS-CoV-2 DC vaccines are being developed (summarized in Table 1). 

Currently, the US biopharmaceutical company Aivita Biomedical plans to conduct a Phase I–II clinical trial of DC vaccines for COVID-19 prevention that are based on autologous monocytes (moDCs) that are loaded with recombinant spike (S)-proteins of COVID-19 without an adjuvant, or in the presence of GM-CSF as an adjuvant [27]. The target population for immunization is the elderly and risk groups of severe disease. After enrolling for screening, the subjects will undergo a nasal swab test to exclude active COVID-19 infection and a rapid test for anti-coronavirus antibodies to exclude pre-existing anti-SARS-CoV-2 antibodies. A total of 50 mL of blood will be collected. Autologous peripheral blood monocytes will be isolated and differentiated into DCs before incubation with SARS-CoV-2 S-protein, during which time the protein is digested into 9 to 25 amino acid peptide sequences that are presented on the dendrites of DCs in conjunction with histocompatibility class I and class II molecules. Only one dose of the vaccine is planned. GM-CSF added together with the DC vaccine is considered to be a safe adjuvant. Strong adjuvants are the so-called pathogen-associated molecular patterns (PAMPs) of the lipopolysaccharide (LPS) type, or pro-inflammatory cytokines that are similar to tumor necrosis factor (TNF)-α. Such adjuvants are often used to mature DCs and endow them high antigen-presenting potential. However, they may be dangerous if they are administered simultaneously with the DC vaccine. 

A resembling Chinese clinical study is already recruiting patients for the immunization with LV-SMENP DC vaccine, which is made by modifying DCs with lentivirus vectors expressing COVID-19 minigene SMENP and immune modulatory genes [28]. Cytotoxic T lymphocytes (CTLs) will be activated ex vivo by LV-DC presenting COVID-19-specific antigens. A participant will receive once LV-DC vaccine (5 × 10^6^ cells) and antigen-specific CTLs (10^8^ cells) via sub-cutaneous injection and intravenous infusion, respectively. This study enrolls both healthy volunteers and COVD-19-infected patients. Lentiviral vectors are modified viruses and, therefore, possess their own adjuvant properties as DCs are readily activated by different viral PAMPs (surface glycoproteins, DNA, and RNA species) via Toll-like (TLRs) and other receptors.

Zhou et al. suggested large-sized graphene oxide (L-GO) nanosheets as a promising adjuvant for DCs that are loaded with SARS-CoV-2 S-protein [29]. L-GO facilitates the aggregation of DC-T-cell clusters to produce a stable microenvironment for T-cell activation. Mice were injected with two doses of DC vaccines (at a dose of 2 × 10^6^ per mouse) at a one-week interval, and three days after the last vaccination, all the immunized mice were intranasally challenged with the adapted SARS-CoV-2. Importantly, L-GO-adjuvanted DCs promoted robust cytotoxic T-cell immune responses against SARS-CoV-2, leading to >99.7% viral RNA clearance in mice that were infected with a clinically isolated SARS-CoV-2 strain. 

Saadeldin suggested an interesting approach that simultaneously targets a tumor and SARS-CoV-2 [11]. Cancer patients would especially benefit from such a vaccine as they are a risk group for severe COVID-19 and need efficient therapy of a malignant disease. 

Important information may be learnt from the already existing studies on anti-infection DC vaccines. DC vaccines have been studied for the treatment of dangerous viral infections since at least 2004, when Reuter et al. showed that vaccination with a DC vaccine that was loaded ex vivo with a viral antigen could protect mice from subsequent infection with the Friend retrovirus (FV) [30]. All the control mice that received DCs without the antigen developed progressive leukemia after FV challenge. In contrast, five of the 14 vaccinated animals were protected against infection, three recovered from FV-induced disease, and only six progressed to lethal leukemia. Virus-specific antibody responses were not induced by the DC vaccination. In contrast, protection correlated with a vaccine-induced CD8^+^ T-cell response that was directed against an immunodominant epitope of FV. CD8^+^ T-cells were critical for the protective effect of the DC vaccine, since in vivo depletion of these cells from immunized mice prevented their protection. Thus, the results demonstrated that antigen-loaded DCs could induce specific cellular immune responses and prevent the retrovirus-induced disease.

Multiple studies demonstrated the safety and induction of an antiviral response in therapeutic models of DC vaccines in HIV patients, nonetheless, DC vaccines are superior in prophylactic usage [31,32]. There are multiple clinical trials and pre-clinical research in the field. The data are thoroughly summarized in the review by Mohamed et al. [32]. Most of the research groups report on the enhancement of specific T-cell responses towards HIV after the immunization. However, some of the groups denote that there are no anti-viral effects and no correlation of the T-cell response and reduced viral load. The studies explore different types of viral antigens (peptides, inactivated viruses, RNA) and routes of their delivery into DCs. Electroporation is an effective way to load DCs with RNA coding for viral proteins. However, viral vectors (adenovirus, poxviruses, lentivirus) may be more effective in pro-inflammatory cytokine promotion and the induction of a CTL response [32]. Elaborate methods are applied in order to enhance the efficiency of therapeutic DC vaccines. Norton et al. investigated the simultaneous expression of HIV-1 antigen with CD40 ligand (CD40L) and a soluble, high-affinity programmed cell death 1 (PD-1) dimer [33]. CD40L activates DCs, whereas PD-1 binds programmed death ligand 1 (PD-L1) to prevent checkpoint activation and strengthen the CTL response. Upon HIV-1 challenge of vaccinated mice, the viral load was significantly suppressed by the approach. The viral mimic, polyinosinic-polycytidylic acid-poly-l-lysine carboxymethylcellulose (Poly-ICLC) is suggested as a potent adjuvant for DC activation during DC load with the viral antigens [34]. 

Research was carried out to generate DC vaccines against hepatitis C virus (HCV) [35,36,37]. Hong et al. developed a super DC stimulant that consists of a modified, secretory TLR-5 ligand and an inhibitor of the negative regulator, suppressor of cytokine signaling-1 (SOCS1) [35]. The agonist consisted of the bacterial filament protein flagellin that activates APCs via its interaction with the surface TLR5 coupled with a small inhibitory RNA of SOCS1. They found that expressing the super stimulant in DCs is drastically more potent and persistent than using the commonly used DC stimuli to enhance the level and duration of inflammatory cytokine production by both murine and human DCs. Moreover, the DCs expressing the super stimulant were more potent to provoke both cellular and humoral immune responses against the HCV antigen in vivo in a murine model. Authors of another study aimed to develop a DC vaccine encoding HCV multiple CTL epitopes that can stimulate T-cell responses in vitro and can be used for immunization in vivo [36]. DCs were infected with recombinant replication-defective adenoviruses expressing 2 HCV sequences that were fused with green fluorescent protein (GFP) and FLAG tags. The recombinant adenoviruses-expressing multiple CTL HCV epitopes effectively infected the DCs in vitro and promoted T-cell antiviral immune responses in vitro, as the CTLs stimulated by the Ad-infected DCs specifically killed Huh7.5 human hepatoma cells. Mekonnen et al. developed another approach to enhance anti HCV DC vaccine efficiency [37]. They suggest necrotic DCs expressing HCV antigen for immunization. They produced a DC2.4 cell line that stably expressed HCV non-structural protein (NS)3–NS3 DC. The vaccination of mice with necrotic NS3 DCs increased the breadth of T-cell responses and enhanced the production of IL-2, TNF-α, and IFN-γ by effector memory CD4^+^ and CD8^+^ T-cells, compared to mice that were vaccinated with live NS3 DC. Necrotic NS3 DC vaccination resulted in enhanced clearance of NS3-positive hepatocytes from the livers of vaccinated mice. 

There are two Phase I/II clinical studies of therapeutic DC-vaccines for treatment of chronic hepatitis C that are listed in the databases of ClinicalTrials.gov. The first one employed autologous DCs that were generated from monocytes in the presence of IFN-α/GM-CSF and pulsed with recombinant HCV Core (1–120) and NS3 (1192–1457) proteins [38]. It was conducted in the Russian Federation and involved 10 patients from 18 to 65 years old. The patients were vaccinated several times via subcutaneous injection of autologous DCs (5 × 10^6^) combined with adjuvant subcutaneous injection of recombinant hIL-2 (250,000 IU). There were no participants with severe adverse reactions and/or with abnormal clinical laboratory values that are related to treatment. Of the 10 participants, four had virological response. A virological response in patients receiving DC-vaccinations is defined as change from baseline in HCV RNA viral load by at least 1 log at 2, 7, and 13 months after the first vaccination. Another study, which was conducted in Spain, aimed to assess the efficacy of therapeutic vaccination in genotype 1 HCV patients using autologous DC that was transduced with a recombinant adenovirus encoding NS3 [39]. Unfortunately, no results were posted.

Chronic hepatitis B (CHB) is another infectious disease that requires the improvement of the natural immune response. Chen et al. showed that an autologous ex vivo-generated DC-vaccine can effectively suppress hepatitis B virus (HBV) replication and reduce the virus load in sera [40]. A Chinese Phase I/II clinical study listed in ClinicalTrials.gov data is searching to provide additional help for CHB patients through hepatitis B vaccine-activated DCs [41]. The anti-virus effects are not satisfying in some of the CHB patients who have been on anti-HBV drugs therapy. DCs are crucial in HBV-specific immunity in the process of priming and boosting HBV-specific CTL and specific T helper (Th) response, however they are defective in CHB patients. Therefore, the authors of the study hope to improve anti-HBV efficacy of the adaptive anti-HBV immune response via ex vivo-generated and reinfused DCs. The preliminary study results showed that the DC-vaccination effectively reconstructed the immunity and elicited virological, serological, and biochemical improvements in some patients with chronic HBV [42,43]. No side effects were observed. Another study suggests that the combination of HBV-pulsed autologous DCs and entecavir could be therapeutically advantageous for patients with CHB [44]. As HBV is the major cause of hepatocarcinoma development, anti-HBV DC vaccines may serve as an anti-cancer therapy in this condition. DCs after the transduction with a recombinant adenovirus containing HBV surface antigen (HBsAg) gene (AdVHBsAg) primed a specific T-cell response in vitro [45]. DCs that were pulsed with hepatitis B virus antigens elicited effective antitumor CTL response against human liver carcinoma cells in a murine model [46]. The antitumor effect of the DC vaccine was enhanced by the addition of β-glucosylceramide that activated hepatic NKT cells. The authors highlight that the inhibiting effect of the DC vaccination on tumor growth was more significant when it was applied before tumor inoculation. 

Cytomegalovirus (CMV) reactivation remains a major cause of morbidity in patients who undergo allogeneic hematopoietic stem cell transplantation (HSCT), occurring in >60% of patients without antiviral prophylaxis [47]. Van Craenenbroeck et al. showed in a successful pilot clinical trial induction of CMV-specific T-cells by RNA-transfected DCs [48]. The DC vaccine both primed naïve T-cells against the virus in CMV seronegative persons and activated the specific memory T-cell response in CMV-positive ones. Ma et al. generated an allogenic donor CMV-peptide-specific DC vaccine for HSCT patients in the presence of a proinflammatory cytokine cocktail. TNF-α, IL-1β, prostaglandin E2, and IL-6 were added to mature the monocyte-derived DCs (moDCs) that were loaded with CMV peptide. The DCs were administered simultaneously with CMV-specific T-cells in an attempt to prevent the development of CMV infection in recipients of allogeneic hematopoietic progenitor stem cells [49]. No immediate adverse reactions were noted on DC vaccination or T-cell infusion. All the patients who received DC vaccination and CMV-specific T-cell infusion demonstrated immune reconstitution against CMV. A Stage I clinical study on CMV-specific DCs has been recently completed in the USA [50]. CMV RNA-pulsed DCs (CMV-DCs) were injected with tetanus toxoid (Td) preconditioning and GM-CSF adjuvant in children and young adults up to 35 years old with WHO grade IV glioma, recurrent malignant glioma, or recurrent medulloblastoma. This study enrolled 10 patients. No severe adverse side effects of the treatment were observed. 

Noticeably, the studies in anti-CMV DC-vaccines suggest an appealing approach that might be utilized for antiviral protection of HSCT-patients who are especially susceptible to any infection, not-mentioning the highly contagious SARS-CoV-2. HLA-matched HSCT donor monocytes might be utilized for the production of any antiviral-DC vaccine including anti-SARC-CoV-2, CMV, etc. and provide protection for the patients via the priming of anti-viral response by the donor T-cells.

Some researchers are studying antifungal DC vaccines against highly virulent fungi, such as *Cryptococcus gattii* that may cause lethal infection in immunocompetent humans [51,52,53,54]. Importantly, DCs that were pulsed with a peptide from *Paracoccidioides brasiliensis* proved to be effective as a therapy in combatting pre-existing fungal disease in an immunocompromised mice model [54]. 

Thus, a range of anti-infectious DC vaccines are currently being developed to treat dangerous and hardly curable chronic diseases. DC vaccines are the most effective for disease prophylaxis. However, therapeutic DC vaccines were shown to provide benefits even in chronic infectious diseases. That is why DC vaccines are expected to be beneficial for anti-SARS-CoV-2 immunization of immunocompromised patients. Different DC maturation stimuli and adjuvants for immunization are still being investigated to develop the most effective and safest DC vaccines. NCT04386252 clinical study intends to use SARS-CoV-2 S-protein to load DCs [27]. However, the authors do not provide any information regarding maturation stimuli, especially required if the protein antigens are used for this purpose. Poly-ICLC, proinflammatory cytokine cocktail (TNF-α, IL-1β, prostaglandin E2, and IL-6), L-GO [29], or other factors that are described above in the review for different antiviral DC vaccines may be added together with peptide or protein antigens to induce DC maturation and enhance their antigen-presenting potential. However, viral vectors may be superior to protein, peptide, and RNA antigens based on the lessons that have been learnt from anti-HIV DC research. Therefore, a DC vaccine from NCT04276896 clinical trial that is based on lentivirus vectors expressing COVID-19 minigene does not need extra maturation factors [28]. Moreover, viral vector-based antigens are expected to induce the most suitable DC-phenotype for the priming of antiviral immunity. Besides, the authors of the study included immune modulatory genes in the construct for enhancement of the DC-vaccine potential. Such an approach was employed in anti-HIV DC-vaccines [33]. However, safety issues of viral vectors are to be thoroughly addressed. It is important to test other viral vectors, such as adenovirus and poxviruses, etc., for the production of anti-SARS-CoV-2 DC vaccines. 

It remains an intriguing and completely unexplored possibility if tolerogenic DC-vaccine may be used to relieve inadequate and excessive immune reactions in the form of cytokine release syndrome (CRS) during COVID-19. Such DCs may be generated in the presence of transforming growth factor β1 (TGF-β1) along with/or 1,25-dihydroxyvitamin D(3) alone [12,13].

DC vaccines might be adapted for SARS-CoV-2 and any infectious disease prophylaxis. Such vaccines are highly desirable for immunocompromised patients. Figure 1 depicts the possible workflow in the generation of a protective anti-SARS-CoV-2 DC vaccine. 

### 2.2. CAR-Effector Cell Therapy 

Exquisite CAR-technology is being intensively exploited to develop new anti-SARS-CoV-2 therapies. Most of the approaches target the surface spike (S) protein of the virus. It is a glycosylated protein that enables viral attachment and cell entry and plays a critical role in the elicitation of the host immune response [55,56]. It binds to the human angiotensin-converting enzyme 2 (ACE2) [57,58]. The S-protein consists of a trimer where each monomer has two subunits (S1 and S2) that are separated by a cleavage site that is recognized by host cell proteases [59]. The S1 subunit is composed of the signal peptide (SP), N terminal domain (NTD), and receptor-binding domain (RBD), whereas the S2 subunit mediates membrane fusion. The S-protein is targeted by neutralizing antibodies [60,61]. At the moment, a number of studies are being performed using traditional CARs that are based on single-stranded genetically engineered antigen-binding sites of antibodies (scFv) that are obtained on the basis of antiviral antibodies to the S-protein of SARS-CoV-2. Antiviral scFv domains were combined with intracellular signaling sequences of the T-cell receptor and co-receptors to form chimeric antiviral CAR receptors to modify immune effectors. 

S1-protein-directed CAR-T cells (Table 2) showed potent in vitro killing of target cells that were loaded with receptor-binding domain (RBD), S1 peptide, or expressing the S1 protein [62]. The CAR included the CR3022-antibody-based scFv followed by a Flag-tag sequence (DYKDDDDK), varying hinge regions, CD28 transmembrane (TM), and intracellular domain and the CD3ζ intracellular domain. A neutralizing antibody from a convalescent SARS patient termed CR3022 binds to the RBD region of both SARS-CoV-1 and SARS-CoV-2 S protein [63]. Thus, the S1-epitope that is bound by CR3022 seems to be highly conserved and might be stable during the evolution of the virus. CAR-T-cell recognition of the SARS-CoV-2 RBD peptide induced the increased expression of the activation antigen CD69 and interferon-γ, granzyme B, perforin, and Fas-ligand on CAR-T cells. The efficacy of killing varied with different sized hinge regions, whereas time-lapse microscopy showed CAR-T-cell cluster formation around the RBD-expressing targets. Cytolysis of the targets was mediated primarily by the granzyme-B/perforin pathway. Moreover, the authors showed in vivo killing of S1-expressing cells by the SARS-CoV-2-directed CAR-T cells in mice. 

However, CAR-T-cells are potent inducers of CRS, which is one of the major COVID-19 pathogenesis factors [64]. Thus, other sources for effector cells that may serve as effective CAR-effectors are being intensively investigated. NK-cells are highly promising antiviral effectors. NK-cells recognize SARS-CoV-2-infected cells via their own innate receptors and kill the infected cells early in the immune response [65]. CAR-modified NK-cells cannot induce graft-versus-host reactions as CAR-T cells do [66,67]. NK-cells are expected by some researchers to be less likely to induce CRS that could potentially exacerbate COVID-19 symptoms [68]. Importantly, CAR-transfected NK-cells may be prepared in advance as an off-the-shelf cellular product without the need for genetic manipulations [69]. However, it should be stressed that use of NK-cells in cellular therapies do not exclude the development of host-versus-graft reactions. It is esential to notice that T-cells may be prepared as an off-the-shelf product as well. Strategies to obtain off-the-shelf T cell biomedical products are reviewed by Mo et al. [70]. 

**Table 2 biomedicines-10-00868-t002:** SARS-CoV-2-targeting CAR-based cellular therapies.

CAR Details, Reference	Target Antigen	Effector Cells	Stage of the Study
scFv-based CR3022 2nd generation CAR, CD28, and CD3ζ activating domains [62]	RBD epitope	T-cells	Pre-clinical research
scFv-based CR3022, 3rd generation CAR, CD28, 4-1BB, and CD3ζ activating domains [71,72,73]	RBD epitope	NK-cells	Pre-clinical research
scFv-based S309, 3rd generation CAR, CD28, 4-1BB, and CD3ζ activating domains [72,73]	RBD epitope	NK-cells	Pre-clinical research
scFv-basedCR3022, MERTK, or MEGF10, or FcRγ(FCER1G), or CD3ζ activating domains [74]	RBD epitope	macrophages	Pre-clinical research
H84T-Banana Lectin (BanLec), 2nd generation CAR, 4-1BB, and CD3 ζ activating domains [75]	high mannose glycosites that decorate viral envelopes	NK-cells	Pre-clinical research
NKG2D-ACE2 CAR-NK cells for therapy of COVID-19, NCT04324996 [76]	S	NK-cells	Phase I/II clinical trial

Therefore, Ma et al. also selected CR3022-antibody for their CAR receptor domain (Table 2), but NK-cells were chosen as effector cells [71]. CR3022-CAR-NK cells can specifically bind to RBD of SARS-CoV-2 and pseudotyped SARS-CoV-2 S protein, and can be activated by pseudotyped SARS-CoV-2-S viral particles in vitro. Further, CR3022-CAR-NK cells can specifically kill pseudo-SARS-CoV-2-infected target cells. In their further work, the scientific team improved their approach for the generation of CAR-NK cells for targeting SARS-CoV-2 and its D614G mutant [72]. CAR-NK cells were generated using the scFv domain of S309 (henceforward, S309-CAR-NK) (Table 2). S309 is a SARS-CoV and SARS-CoV-2 neutralizing antibody that targets the highly conserved region of the SARS-CoV-2 S glycoprotein, therefore it would be more likely to recognize different variants of SARS-CoV-2 isolates [77]. S309-CAR-NK cells can specifically bind to pseudotyped SARS-CoV-2 virus and its D614G, N501Y, and E484K mutants. Furthermore, S309-CAR-NK cells could specifically kill target cells expressing SARS-CoV-2 S protein in vitro and exhibited superior killing activity and cytokine production, compared to that of the CR3022-CAR-NK cells.

An interesting approach was suggested by Fu et al. [74]. Their team genetically armed human macrophages with CARs to reprogram their phagocytic activity against SARS-CoV-2 (Table 2). After investigation of CAR constructs with different intracellular receptor domains, they found that, although cytosolic domains from MER proto-oncogene tyrosine kinase (MERTK) (CAR_MERTK_) did not trigger antigen-specific cellular phagocytosis or killing effects, unlike those from MEGF10, FcRγ, and CD3ζ did, these CARs all mediated similar SARS-CoV-2 clearance in vitro. Notably, they reported that a CAR with the intracellular domain of MERTK did not show a notable killing effect in antigen-expressing cell-based models compared with other CARs but did demonstrate antigen-specific clearance of SARS-CoV-2 virions in vitro without the secretion of proinflammatory cytokines These results suggested that CAR_MERTK_ drove an ‘immunologically silent’ scavenger effect in macrophages and paved the way for further investigation of CARs for the treatment of individuals with COVID-19, particularly those with severe cases at a high risk of hyperinflammation. 

A lectin from banana (BanLec) is suggested as a potential receptor of a SARS-CoV-2-targeting CAR [75]. It binds high mannose glycans on viral envelopes, exerting an anti-viral effect. A point mutation (H84T) divorces BanLec mitogenicity from antiviral activity. SARS-CoV-2 contains high mannose glycosites in proximity to the RBD of the envelope S protein. H84T-BanLec CAR-NK cells (Table 2) reduced S-protein pseudotyped lentiviral infection of 293T cells expressing ACE2, the receptor for SARS-CoV-2. NK-cells were activated to secrete inflammatory cytokines when they were in culture with virally-infected cells. Such an approach is advantageous as it is not sensitive to mutations in different virus strains. However, the specificity of the CAR has to be thoroughly evaluated. 

At the moment, a Phase I/II clinical trial NCT04324996 (Table 2) is being carried out in China for the treatment of COVID-19 using allogeneic CAR-NK [76]. The authors are testing genetically modified primary human NK-cells as a therapeutic BMCP, which differ favorably from T lymphocytes. Such an allogeneic product is practically universal and can be administered in case of emergency to any recipient. Therefore, the authors suggest developing such a product, cryopreserve it, and store it until it is required. The authors modified NK-cells with ACE 2 receptor binding to S-protein to recognize the virus and infected cells. Self NK receptors can work as additional means of virus recognition, while normally they serve to detect cells that are infected with viruses. The authors focus on the activation receptor NKG2D. The innate antiviral NK-cell functions give them significant advantages as antiviral effectors for CAR modification. Additionally, the NK-cells were modified to secrete the IL-15 super agonist (a chimeric soluble protein consisting of IL-15 and the alpha chain of its receptor sIL-15/IL-15Rα). According to the authors, this modification serves to increase the lifetime of genetically modified NK-cells in a host body. To reduce the CRS another modification was made in the study, the tested CAR-NK cells had a constructed single-stranded variant of the scFv to GM-CSF. 

Mathematical models show the potential efficacy of COVID-19-specific CAR-T cells in the anti-viral treatment [78,79].

Zhu et al. investigated the anti-viral potential of programmed nanovesicles (NV) that were derived from bispecific CAR-T cells that simultaneously express two different scFv domains, named CR3022 and B38, to target SARS-CoV-2 [80]. NV that expresses both CR3022 and B38 (CR3022/B38 NVs) have a stronger ability to neutralize S-pseudovirus infectivity than nanovesicles that express either CR3022 or B38 alone. Notably, the co-expression of CR3022 and B38, which target different epitopes of S-protein could reduce the incidence of viral resistance. Remdesivir was encapsulated in CR3022/B38 NVs by electroporation and then delivered into the infectious sites of SARS-CoV-2 based on CR3022/B38 NVs targeting. The authors propose that CR3022/B38 NVs have the potential ability to target antiviral drugs to the main site of viral infection, thereby enhancing the antiviral ability by inhibiting intracellular viral replication and reducing adverse drug reactions.

CAR-T cells were fruitfully tested for the treatment of several viral diseases, such as human CMV (HCMV), HIV, Epstein-Barr Virus EBV, HBV, and HCV [81,82]. HCMV-directed CARs were reviewed by Bednar and Ensser [81]. HCMV-targeting CAR-T cells were efficient in the lysis of the infected cells in vitro, and secreted pro-inflammatory cytokines on contact with their targets (TNF and IFN-γ). However, some of the researchers report on a low cytolytic activity of the CAR-lymphocytes. It was observed that HCMV-infected cells can resist cytotoxic lysis by CAR+ T-cells presumably via viral effector proteins. The efficiency of HCMV-targeting CARs was proven in a mouse model as well. 

Apart from HCMV, another clinically relevant member of the herpesviruses is EBV. EBV is a latent and oncogenic human herpesvirus. EBV expresses an abundance of the viral glycoprotein gp350 on the surface of cells during lytic infection, which makes this glycoprotein a potential target for CAR^+^ T-cell therapy. A second generation CAR was created by Slabik et al. based on two highly neutralizing antibodies targeting gp350 (7A1 and 6G4) [83]. The scFv regions were fused to CD28/CD3ζ signaling domains in a retroviral vector. The produced gp350CAR-T cells specifically recognized and killed gp350^+^ 293T cells in vitro. The best-performing 7A1-gp350CAR-T cells were cytotoxic against the EBV^+^ B95-8 cell line, showing selectivity against gp350^+^ cells. Fully humanized Nod.Rag.Gamma mice that were transplanted with cord blood CD34^+^ cells and infected with the EBV/M81/fLuc lytic strain were monitored dynamically for viral spread. The infected mice recapitulated EBV-induced lymphoproliferation, tumor development, and systemic inflammation. The authors tested adoptive transfer of autologous CD8^+^ gp350CAR-T cells that were administered protectively or therapeutically. After gp350CAR-T cell therapy, 75% of mice controlled or reduced EBV spread and showed lower frequencies of EBER^+^ B cell malignant lymphoproliferation, lack of tumor development, and reduced inflammation. The CAR-T cells targeting EBV LMP-1 showed potency in vitro and in mice that were xenografted with an EBV-negative nasopharyngeal carcinoma cell line (SUNE1) that was genetically modified for LMP1 overexpression [84]. The viral LMP1 is expressed in most EBV-associated lymphoproliferative diseases and malignancies, and critically contributes to pathogenesis and disease phenotypes [85]. The early Phase I NCT04657965 clinical trial (China, not yet recruiting) intends to investigate anti-LMP1 CAR T-cells for the treatment of EBV infection and EBV-caused hematological malignancies [86]. 

CAR-T cells with moderate effectiveness have been used in the treatment of a serious viral disease that is associated with HIV [81,87]. Between 1995 and 2005, several clinical trials investigated the safety and efficacy of using CD4ζCAR-T cells in HIV-infected individuals, reviewed in [87]. The outcomes of these studies reinforced the safety and feasibility of ex vivo adoptive T-cell gene therapy, but ultimately, treatment failed to durably reduce the viral burden within blood and tissue reservoirs. The first anti-HIV-CARs were based on the CD4-domain, which binds HIV, and were first generation CARs with only one activating domain from CD3. To improve the activity and persistence of CAR-T cells, second and third-generation anti-HIV-CARs were designed. Compared to the first generation CAR-T cells, the second-generation CAR-T cells were more potent in suppressing HIV replication in vitro. Furthermore, in a humanized mouse model of HIV infection, they preserved CD4^+^ T cell counts, reduced HIV burden, and expanded to a greater extent than the first-generation CAR-T cells. Still, CAR receptors that were built on CD4^+^ T elements have been shown to make CAR-T cells susceptible to HIV infection. Thus, several ways to protect such CAR-T-cells from viral entry were designed. For example, bispecific CAR in which a CD4 segment is linked to a scFv of the 17b human monoclonal antibody recognizing a highly conserved CD4-induced epitope on gp120 that is involved in coreceptor binding protects CD8-CAR-T-cells from viral entry and makes them even more efficient in the infected cell lysis [88]. Another way to overcome this potentially harmful situation is to equip the CD4ζCAR system with a viral fusion inhibitor (C46 peptide) or small hairpin RNAs to break down the HIV-1 coreceptor (CCR5) and degrade the viral RNA [89].

Several groups have explored targeting HIV-infected cells using second-generation CARs with alternative antigen-binding moieties [87]. CARs containing scFv sequences that are derived from broadly neutralizing antibodies (bNAbs) have been constructed that target conserved sites within the Env protein, including the CD4-binding site, the gp41 membrane-proximal external region, and variable region glycans. Despite the antiviral capacity of scFv-based CAR-T cells in vitro, several factors may limit their therapeutic potential in humans. To become a broadly applicable therapy, scFv-based CAR-T cells must overcome HIV escape, be effective against the diversity of HIV strains, and be non-immunogenic so that they can persist for decades.

Several approaches are currently being evaluated to increase efficiency of anti-HIV-CARs. The PD-1 immune checkpoint blockade was shown to be an effective way to enhance anti-HIV-CAR activity in vitro and in a mouse model as anti-HIV-CAR lymphocytes are subjected to significant exhaustion due to chronic infection [90]. 

During chronic HIV or simian immunodeficiency virus (SIV) infection prior to AIDS progression, the vast majority of viral replication is concentrated within B-cell follicles of secondary lymphoid tissues. Thus, Pampusch et al. investigated whether the infusion of T-cells expressing a SIV-specific CAR and the follicular homing receptor, CXCR5, could successfully kill viral-RNA+ cells in targeted lymphoid follicles in SIV-infected rhesus macaques [91]. In this study, CD4 and CD8 T-cells from rhesus macaques were genetically modified to express antiviral CAR and CXCR5 moieties (generating CAR/CXCR5-T cells) and autologously infused into a chronically infected animal. These CAR/CXCR5-T cells replicated in vivo within both the extrafollicular and the follicular regions of lymph nodes and accumulated within lymphoid follicles. Overall, CAR/CXCR5-T cell-treated animals maintained lower viral loads and follicular viral RNA levels than the untreated control animals, and no outstanding adverse reactions were noted. These findings indicate that CAR/CXCR5-T cell treatment is safe and holds promise as a future treatment for the durable remission of HIV.

The fusion inhibitor C34 linked to CXCR4 is potent and interferes with the entry of diverse HIV strains regardless of their tropism [92]. Recently such a fusion inhibitor was employed to generate HIV-resistant and HIV-specific CAR-modified CD4^+^ T-cells (CAR_4_) [93]. The authors demonstrated that CAR_4_ T-cells directly suppressed in vitro HIV replication and eliminated virus-infected cells. Notably, CAR_4_ T-cells containing intracellular domains (ICDs) that were derived from the CD28 receptor family (ICOS and CD28) exhibited superior effector functions compared to the tumor necrosis factor receptor (TNFR) family ICDs (CD27, OX40, and 4-1BB). However, despite demonstrating limited in vitro efficacy HIV-resistant CAR_4_ T-cells expressing the 4-1BBζ ICD exhibited profound expansion, concomitant with reduced rebound viremia after antiretroviral therapy (ART) cessation and protection of CD4^+^ T-cells (CAR^-^) from HIV-induced depletion in humanized mice. Moreover, CAR_4_ T-cells enhanced the in vivo persistence and efficacy of HIV-specific CAR-modified CD8^+^ T-cells that were expressing the CD28ζ ICD, which alone exhibited poor survival. Collectively, these studies demonstrate that HIV-resistant CAR_4_ T-cells can directly control HIV replication and augment the virus-specific CD8^+^ T-cell response, highlighting the therapeutic potential of engineered CD4^+^ T-cells to engender a functional HIV cure.

The current Phase 1 clinical study of anti-HIV-CAR-T cells NCT03617198 (USA) is based on a second generation CAR (41B-B, CD3ζ ICD) with CD4-receptor part [94,95]. The anti-viral protection of the CAR is provided by the system that is described by Liu et al. [88]. Currently, there are several more on-going clinical trials of anti-HIV-CARs: NCT04648046 (USA) [96], NCT04863066 (China, not yet recruiting) [97], NCT03240328 (China) [98].

CARs are also being tested to restore/enhance HBV-specific immune responses in chronic hepatitis B [99]. CAR that is directed against the HBV surface proteins enabled human T-cells to kill HBV-infected human hepatocytes and to eliminate viral covalently closed circular DNA (cccDNA) in vitro [100]. Adoptively transferred T-cells retained their function in vivo and controlled virus replication without significant T-cell–related toxicity in a model of persistent HBV infection in HBV transgenic (HBVtg) mice with a functional immune system [101]. 

Kruse et al. generated a novel CAR targeting HBsAg and evaluated its ability to recognize HBV+ cell lines and HBsAg particles in vitro and tested whether human HBsAg-CAR-T cells would have efficacy against HBV-infected hepatocytes in human liver chimeric mice [102]. HBsAg-CAR-T cells recognized HBV-positive cell lines and HBsAg particles in vitro as judged by cytokine production. However, HBsAg-CAR-T cells did not kill the HBV-positive cell lines in cytotoxicity assays. The adoptive transfer of HBsAg-CAR-T cells into HBV-infected humanized mice resulted in their accumulation within the liver and a significant decrease in plasma HBsAg and HBV-DNA levels compared with the control mice. Notably, the fraction of HBV core-positive hepatocytes among total human hepatocytes was greatly reduced after HBsAg-CAR-T cell treatment, pointing to noncytopathic viral clearance. 

Anti-HBV CAR-T cell therapy is a promising therapeutic approach for the treatment of chronic hepatitis B and HBV-associated cancer. Due to the high number of target cells, however, side effects such as CRS or hepatotoxicity may limit safety. Klopp et al. created anti-HBV-CAR-T cells with a safeguard mechanism, which allows the depletion of transferred T-cells on demand [103]. In this study, the T-cells were generated by retroviral transduction to express an HBV-specific chimeric antigen receptor (S-CAR), and in addition either inducible caspase 9 (iC9) or herpes simplex virus thymidine kinase (HSV-TK) as a safety switch.

Festag et al. engineered fully human, second-generation CAR-T cells targeting HBsAg and tested them in an AAV-HBV mouse model with specific tolerance to human HBsAg-CAR [104]. In this system, long-lasting antiviral effects were demonstrated with 2 log10 decrease of HBsAg and 60% reduction of HBV-DNA for up to 110 days upon adoptive transfer. However, HBsAg-CAR-T cells failed to completely clear HBV in the animals. 

Sautto et al. designed the first CARs against HCV targeting the HCV/E2 glycoprotein (HCV/E2) [105]. Anti-HCV/E2 CARs were composed of a scFv that was obtained from a broadly cross-reactive and cross-neutralizing human monoclonal antibody (mAb), e137, that was fused to the intracellular signaling motif of the costimulatory CD28 molecule and the CD3ζ domain. The activity of the CAR-grafted T-cells was evaluated in vitro against HCV/E2-transfected cells as well as hepatocytes that were infected with cell culture-derived HCV (HCVcc). The T-cells expressing anti-HCV/E2 CARs were endowed with specific antigen recognition that was accompanied by degranulation and secretion of proinflammatory and antiviral cytokines, such as IFN-γ, IL-2, and TNF-α. Moreover, the CAR-grafted T-cells were capable of lysing the target cells of both hepatic and non-hepatic origin expressing on their surface the HCV/E2 glycoproteins of the most clinically relevant genotypes, including 1a, 1b, 2a, 3a, 4, and 5. Finally, and more importantly, they were capable of lysing HCVcc-infected hepatocytes.

Thus, most of the antiviral CAR studies are at the stage of pre-clinical research. All of them, except for SARS-CoV-2, deal with chronic diseases. That is why many of the scientists are faced with viral mechanisms that enable viruses to escape natural human immunity. Such viral properties may dampen CAR-lymphocyte efficiency too. Unfortunately, most of these mechanisms remain to be elucidated. COVID-19 is an acute infection, however, SARS-CoV-2 certainly possesses certain properties enabling it to weaken natural adaptive immunity. These peculiarities are still to be discovered and might be crucial for successful antiviral treatment. There are several well-established strategies for the design of anti-viral CARs that may be adapted for the generation of effective anti-SARS-CoV-2 CARs. The most detailed information is available from the vast and well-developed anti-HIV-CAR research field. These studies clearly indicate that second generation CARs are superior in antiviral protection to first generation CARs [81,86,87]. Moreover, TNFR family activating domains (CD27, OX40, and 4-1BB) seem to be important for an efficient in vivo functioning of CAR-modified cells, providing cell expansion [93]. Whereas activating domains that are derived from the CD28 receptor family (ICOS and CD28) are important for effector functions. Thus, third generation SARS-CoV-2-directed CARsbased on CD28-, 4-1BB-, and CD3ζ-activating domains are expected to be highly efficient in viral clearance [71,72,73]. Targeting individual viral epitopes by scFv-based CARs have a disadvantage of possible viral escape via its rapid mutations. Therefore, CARs that are employing other receptor domains with broader specificity such as ACE2 or BanLec seem preferential in this aspect. However, possible cross-reactivity of such CARs with other ligands may become a problem. ACE2 is a receptor for angiotensin, it is an integral part of the renin–angiotensin–aldosterone system that exists to keep the body’s blood pressure in check. Thus, ACE2-bearing CAR-lymphocytes may alter the natural control of blood pressure. Direct comparison of different CARs’ efficiency and safety is required to choose the best variants. Protection of anti-SARS-CoV-2 CAR-lymphocytes from the viral entry and infection might be required just as in the case of HIV. Potentially, such methods may be developed based on the previous research of anti-HIV CAR-lymphocytes. Additional mechanisms to reduce or turn off undesired CRS (suicide genes, inflammatory cytokine-neutralizing scFv) are highly desirable for safety reasons during CAR-therapy of COVID-19 [76,104].

The possible CAR-based strategy for COVID-19 treatment is presented in Figure 2.

### 2.3. Potential Advantages of Universal CAR Technology in Anti-SARS-CoV-2 Treatment 

NK-cells or lymphocytes that are devoid of αβ-TCR possess appealing properties of the potential use of allogeneic material as universal off-the-shelf product for any individual on demand [24]. However, conventional CARs are directed towards just one protein epitope. That is a huge disadvantage especially when they are used in antiviral therapy due to the high viral variability. The technique of so called universal CARs(unCAR) that are based on the antibodies and their receptors may overcome the problem (Figure 3). 

The advantage of this approach is enhanced universality and adaptability of the system. The specificity and affinity of binding to a viral antigen is determined exclusively by antibodies, which are easier to obtain than conventional CAR, and it is possible to replace antibodies with other clones or combine them. The system can use different monoclonal antibodies to different viral epitopes or even mixtures. It is highly probable that convalescent plasma may be used together with unCAR-modified lymphocytes as well.

Actually, there are several types of unCARs. The low-affinity IgG receptor CD16 is often used for the design of such CARs [106,107,108,109]. The high affinity CD64 receptor may also serve as a receptor part of an unCAR [26,109,110]. The action of such CARs resembles ADCC as it was discussed in the Introduction, part 1. 

Other approaches to generate unCAR were established and successfully tested in vitro and in experimental animals. Avidin may be linked to CAR signaling domains and binds biotinylated antibodies [111]. The strategy employs a biotin-binding immune receptor (BBIR) that is composed of an extracellular-modified avidin that is linked to an intracellular T-cell signaling domain. BBIR T-cells recognized and bound exclusively to cancer cells that were pre-targeted with specific biotinylated molecules (antibodies). 

A CAR that binds a fluorescein isothiocyanate (FITC) molecule, termed anti-FITC CAR, efficiently interacts with antibodies that are conjugated with FITC [112]. 

The strategy of universal CARs was tested for anti-HIV treatment [113]. Lim et al. reported the development of universal CAR-NK cells, which recognized 2,4-dinitrophenyl (DNP) and could subsequently be redirected to target various epitopes of gp160 using DNP-conjugated antibodies as adaptor molecules. They showed that such CAR-NK cells could recognize and kill mimic HIV-infected cell lines expressing subtypes B and C gp160. They confirmed that HIV-infected primary human CD4^+^ T-cells could be effectively killed using the same approach. Given that numerous anti-gp160 antibodies with different antigen specificities are readily available, this modular universal CAR-NK cell platform can potentially overcome HIV diversity, thus providing a promising strategy to eradicate HIV-infected cells.

Therefore, the technology of universal antiviral CAR-effectors is beneficial for a number of reasons, including an opportunity of using a cryopreserved donor product for an emergency, employment of different antibodies and their mixtures [24]. Moreover, CARs that are based on CD16 or CD64 are not immunogenic as they are natural biomolecules of human origin.

## 3. Conclusions

Anti-viral BMCP, such as DC vaccines and anti-viral CAR-effectors, are promising in problematic cases of SARS-CoV-2 therapy and prophylaxis. Up-to-date studies in anti-viral DC vaccines and CARs directed against other infections prove this approach to be feasible, safe enough, and efficient.

## Figures and Tables

**Figure 1 biomedicines-10-00868-f001:**
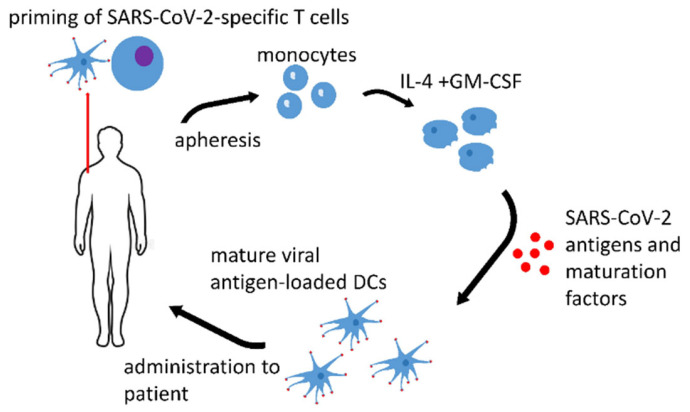
Anti-SARS-CoV-2 DC vaccine. Monocytes are isolated from the autologous apheresis product. Immature DC are generated by the standard procedure in the presence of IL-4 and GM-CSF. The final protective DCs are obtained by the loading of the immature DCs with SARS-CoV-2 antigens in the presence of maturation factors.

**Figure 2 biomedicines-10-00868-f002:**
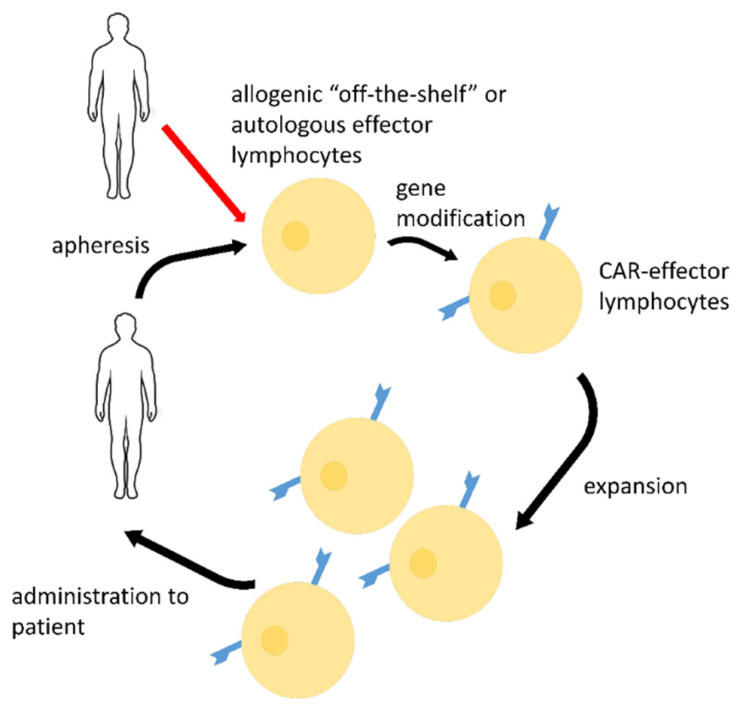
Possible strategy of COVID-19 emergency treatment by SARS-CoV-2-directed CAR-lymphocytes (other effector cells, such as macrophages may be used too). Autologous or allogenic off-the-shelf lymphocytes may be produced from apheresis products via gene modification and further ex vivo expansion. Such CAR-effectors may be stored, cryopreserved, and administered to patients in case of infection.

**Figure 3 biomedicines-10-00868-f003:**
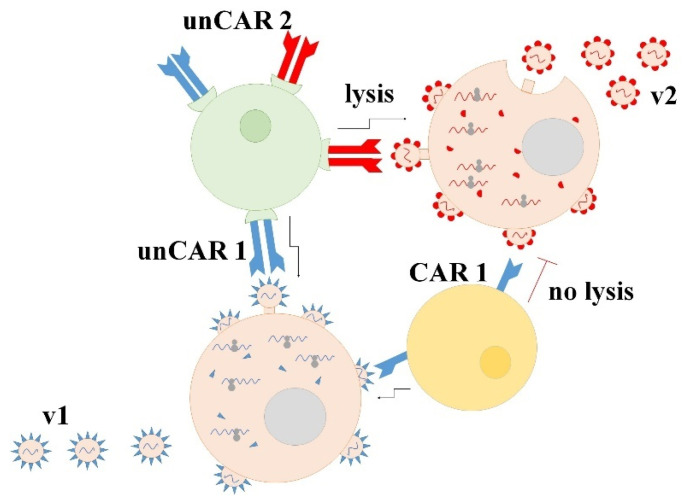
CAR lymphocytes are able to recognize and lyse virus-infected cells. Universal CARs may bind different antibodies. Thus, they may target different viral epitopes. v1—virus strain 1; v2—virus strain 2; unCAR1 is composed of antibodies specific to strain 1 (IgG-s1) bound to IgG-receptor-based CAR, a conventional CAR1- is based on the same IgG-s1; unCAR2 is composed of antibodies that are specific to IgG-s2 that is bound to IgG-receptor-based CAR.

**Table 1 biomedicines-10-00868-t001:** Contemporary anti-SARS-CoV-2 DC-vaccine studies.

Name of the Vaccine, ClinicalTrials.gov Identifier or Authors if Applicable, Reference	Type of the Antigen and Adjuvant	Stage of the Study
AV-COVID-19, NCT04386252 [27]	S-protein with GM-CSF or without an adjuvant	Phase I–II clinical trial, not yet recruiting
LV-SMENP DC, NCT04276896 [28]	lentivirus vectors expressing COVID-19 minigene	Phase I–II clinical trial, recruiting
Zhou et al. [29]	S-protein with graphene oxide (GO) nanosheets	pre-clinical research

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
