# Peer review of "Antiviral Cell Products against COVID-19: Learning Lessons from Previous Research in Anti-Infective Cell-Based Agents"

_biomedicines, 2022, doi:10.3390/biomedicines10040868_

Round 1
Reviewer 1 Report
In this review titled “Antiviral cell products against COVID19”, the authors introduced the major antiviral cell products are being developed against COVID-19, including DC vaccines and CAR-T cells. Furthermore, the authors introduced relevant strategies used for the design of general antiviral cell products. Overall, this is a timely and interesting review. And this manuscript is well written in general. However, certain key information needs to be added so that this review could be more comprehensive and unbiased.
Major issues
- It was nice to introduce the research and design of antiviral cell products against different viruses besides SARS-CoV-2. However, it seems that the major goal of this review is to summarize and guide the research on antiviral cell products against SARS-CoV-2. With this goal in mind, the authors didn’t explicitly summarize what they learned from the research on antiviral cell products against other viruses, which could be applied to the anti-SARS-CoV-2 cell products.
- Is there any research on antiviral cell products against coronaviruses besides SARS-CoV-2? If there is, it could be important to introduce this piece of information too.
- The information in table 1 is not very comprehensive. Some important information could be shown too, such as how many doses (1 dose, 2 doses or multiple doses?) were given in these studies? How many cells were given in each dose?
- Line 146-147, the full name of PAMPs is not accurate, please double check.
- The full name of scFv appeared at least twice in this manuscript (line 101 and line 319), and it was written differently. Please double check the accuracy.
- Line 345- 346, “NK cells are less likely to induce CRS that could potentially exacerbate COVID-19 symptoms [68]. Importantly, CAR-transfected NK cells may be prepared in advance as an off-the-shelf anti-cancer cellular product [69]”, I’m not sure NK cells are superior in these two aspects. First, ref 68 only studied NK cells, and didn’t use T cells, so it didn’t prove NK cells induce less CRS. Second, CAR-T cells can also be made as an off-the-shelf product (Mo F, Mamonkin M, Brenner MK, Heslop HE. Taking T-cell oncotherapy off-the-shelf. Trends in Immunology. 2021 Mar 1;42(3):261-72). Please rewrite this part to ensure the accuracy and scientific soundness.
Author Response
Response to Reviewer 1 Comments
- It was nice to introduce the research and design of antiviral cell products against different viruses besides SARS-CoV-2. However, it seems that the major goal of this review is to summarize and guide the research on antiviral cell products against SARS-CoV-2. With this goal in mind, the authors didn’t explicitly summarize what they learned from the research on antiviral cell products against other viruses, which could be applied to the anti-SARS-CoV-2 cell products.
Response 1: The explicit summary of the important topics learnt from previous studies are added in the end of each section. Please, see the revised manuscript.
- Is there any research on antiviral cell products against coronaviruses besides SARS-CoV-2? If there is, it could be important to introduce this piece of information too.
Response 2: During preparation of the review, I thoroughly searched for any studies dealing with antiviral cell products and have not found any mentions of other coronaviruses besides SARS-CoV-2.
- The information in table 1 is not very comprehensive. Some important information could be shown too, such as how many doses (1 dose, 2 doses or multiple doses?) were given in these studies? How many cells were given in each dose?
Response 3: It is rather complicated to make adequate comparison of the required information in the table, as the studies are quite different. One of the studies is just an experimental work in mice. NCT04386252 clinical study intends to administer DCs generated from 50 ml of a patient’s blood, than a patient of NCT04276896 study is recieving total 5x106 DCs and 1x108 ex vivo generated antigen-specific CTLs. As there are only three such studies, I added the information in the main text. I hope it would be convenient for the readers. Please, track the changes.
- Line 146-147, the full name of PAMPs is not accurate, please double check.
Response 4: The full name of PAMPs was corrected.
- The full name of scFv appeared at least twice in this manuscript (line 101 and line 319), and it was written differently. Please double check the accuracy.
Response 5: The accuracy was checked, the corrections were made.
- Line 345- 346, “NK cells are less likely to induce CRS that could potentially exacerbate COVID-19 symptoms [68]. Importantly, CAR-transfected NK cells may be prepared in advance as an off-the-shelf anti-cancer cellular product [69]”, I’m not sure NK cells are superior in these two aspects. First, ref 68 only studied NK cells, and didn’t use T cells, so it didn’t prove NK cells induce less CRS. Second, CAR-T cells can also be made as an off-the-shelf product (Mo F, Mamonkin M, Brenner MK, Heslop HE. Taking T-cell oncotherapy off-the-shelf. Trends in Immunology. 2021 Mar 1;42(3):261-72). Please rewrite this part to ensure the accuracy and scientific soundness.
Response 6: The part was rewritten, the necessary corrections were made.

Reviewer 2 Report
This review summarizes the DC vaccines and CAR lymphocytes therapeutic potential for COVID-19. Review can be shortened by taking out the unnecessary discussion of these treatment options and their potential in other diseases. Authors need to focus and discuss these treatment options by keeping in mind their importance and correlation in COVID-19 management. Instead of Figure 1, authors can provide a figure summarizing all the relevant details discussed in the review for better understanding and visibility for reader.
Author Response
This review summarizes the DC vaccines and CAR lymphocytes therapeutic potential for COVID-19. Review can be shortened by taking out the unnecessary discussion of these treatment options and their potential in other diseases. Authors need to focus and discuss these treatment options by keeping in mind their importance and correlation in COVID-19 management. Instead of Figure 1, authors can provide a figure summarizing all the relevant details discussed in the review for better understanding and visibility for reader.
I agree that our review contains a lot of information about DC vaccines and CAR lymphocytes in different infectious diseases. However, such data might be of high usefulness for specialists interested in anti-SARS-CoV-2 biomedical cell products. The title of the manuscript is misleading as it focuses only on barely investigated SARS-CoV-2-targeting products. I have changed it. Also, I explicitly summarized what was learned from the research on antiviral cell products against other viruses, which could be applied to the anti-SARS-CoV-2 cell products.
We include two more figures summarizing details of DC-vaccines and CAR-based treatment for COVID-19. Please, see the revised manuscript.

Round 2
Reviewer 1 Report
All concerns resolved